# The Relationship Between School Organizational Climate and Teachers’ Organizational Citizenship Behaviors: The Mediating Role of Teaching Efficacy and Moderating Role of Optimistic Traits

**DOI:** 10.3390/bs14121130

**Published:** 2024-11-25

**Authors:** Wenmei Sun, Xubo Liu, Yiwen Liu, Sasa Ding, Yan Jiang, Ziyan Lv

**Affiliations:** Faculty of Education, Henan Normal University, Xinxiang 453007, China; 19100274046@stu.htu.edu.cn (X.L.); 2210183023@stu.htu.edu.cn (Y.L.); 2310283137@stu.htu.edu.cn (S.D.); 2210283136@stu.htu.edu.cn (Y.J.); 2210283142@stu.htu.edu.cn (Z.L.)

**Keywords:** optimistic traits, school organizational climate, teaching efficacy, teachers, organizational citizenship behaviors

## Abstract

This study examined the relationship between school organizational climate and teachers’ organizational citizenship behavior, as well as the mediating role of teaching efficacy and the moderating role of optimistic traits. This study was based on social information processing theory, resource conservation theory, and the broaden-and-build theory of positive emotions. We conducted a comprehensive survey of 500 educators from Chinese primary and secondary schools using the Organizational Citizenship Behavior Questionnaire, School Organizational Climate Scale, Sense of Teaching Efficacy Scale, and Optimistic Traits Questionnaire as assessment tools. The results demonstrated that (1) school organizational climate significantly and positively predicted teachers’ organizational citizenship behaviors; (2) teaching efficacy partially mediated the relationship between school organizational climate and teachers’ organizational citizenship behaviors; and (3) optimistic traits moderated the second half of the mediation model in which school organizational climate influenced teachers’ organizational citizenship behaviors through teaching efficacy. Our findings indicated that school organizational climate was an important environmental factor affecting teachers’ organizational citizenship behaviors through their sense of teaching efficacy. Optimistic traits had an important protective effect on teachers’ organizational citizenship behaviors.

## 1. Introduction

Educator excellence forms the foundation for the advancement of superior education. Enhancing teacher development is indispensable for elevating educational standards and serves as the bedrock for fostering societal progression and constructing cohesive communities. On 2 April 2022, China’s Ministry of Education and eight additional departments released the New Era Strong Teachers Initiative in Basic Education, which stressed that strengthening the development of the teaching force is a top priority for educational development.

Primary and secondary school teachers are the most important human resources in basic education. Due to the unique characteristics of their work, such as high autonomy, limited control, and implicit rather than explicit work, teachers’ organizational citizenship behaviors, such as dedication, ethical requirements, and pursuit of knowledge, affect school organizational effectiveness. Positive organizational citizenship behavior improves teachers’ morale, stimulates their innovation, and serves as a criterion for measuring the effectiveness of school organizations [1].

Teachers’ organizational citizenship behavior is spontaneous and positive beyond the expectations of the role that teachers demonstrate outside their job responsibilities, which reflects their initiative, responsibility, and altruism [2]. Despite the increasingly refined categorization of dimensions within the research field of organizational citizenship behaviors (OCBs) in recent years, its essence has consistently focused on voluntary behaviors that extend beyond formal job responsibilities [3]. Notably, numerous past studies have treated OCBs as comprehensive concepts for discussion [4,5]. A systematic literature review on OCBs demonstrated that some scholars believe that these behaviors primarily encompass two major dimensions: assistance and support provided to individuals (such as colleagues) and spontaneous adherence to and maintenance of organizational rules [6]. However, when exploring the antecedent variables of OCBs, some scholars integrate these two dimensions and conduct in-depth analysis as a whole concept. In the context of this study, we focus on the teacher population, aiming to explore their positive behaviors displayed beyond fulfilling basic job responsibilities and transcending traditional role definitions. These behaviors not only benefit the organization as a whole but also profoundly impact individual-related aspects. Given this, this study decides to adopt a holistic perspective, treating OCBs as comprehensive concepts for discussion to fully reveal the positive behavioral characteristics exhibited by teachers in their work practices.

Previous studies have demonstrated that organizational citizenship behaviors contribute to job performance, improve organizational survivability and core competitiveness in uncertain environments, enhance team decision-making effectiveness, and promote positive cultural organizations [7,8,9,10,11].

Teachers’ organizational citizenship behaviors strengthen the development of a new era of teachers by improving teaching force quality, advancing the eco-friendly evolution of education, and ensuring the establishment of a robust groundwork for nurturing teaching professionals in the contemporary age. Teachers’ organizational citizenship behaviors involve enthusiasm and dedication to education and boosting the school level [1]. In addition, advocating and practicing organizational citizenship behaviors can stimulate teachers’ professional enthusiasm and innovative spirit and enhance their professional competence and teamwork. This could contribute to constructing an elite and proficient teaching cadre, thereby offering a robust foundation for advancing contemporary education.

A literature review reveals that most studies exhibit a limitation of single-level analysis. By considering them comprehensively, this research aims to further elucidate how environmental factors and individual cognitions influence teachers’ behaviors through individual psychological variables and traits. Additionally, most studies have overlooked the potential role of optimistic traits, which are important psychological traits, in moderating the relationship between organizational climate and teachers’ behaviors. Therefore, this study intends to introduce teaching efficacy as a mediator to uncover how teachers’ perception of school organizational climate affects their organizational citizenship behavior. Furthermore, it explores how variations in teaching efficacy among different teachers influence their behavioral manifestations.

### 1.1. The Relationship Between School Organizational Climate and Teachers’ Organizational Citizenship Behaviors

The organizational climate of a school is not limited to the social interactions between teachers and principals; it encompasses refined organizational behaviors in the school and efficient human resource management strategies, which shape a professional, structured, and dynamic school environment [12]. This is the foundation for teachers’ organizational citizenship behaviors [1].

However, as emphasized by Ghavifekr and Pillai [13], teachers’ perception of the schoolwork environment is central to school organizational climate. The quality of school organizational climate is not an objectively existing and immutable entity; instead, it needs to be embodied and confirmed through teachers’ personal perceptions. In other words, only when teachers truly perceive a positive organizational climate within the school can we say such a climate exists and is effective. This perception is a product of the combination of subjective and objective factors, encompassing not only teachers’ direct experiences of the school’s policies, systems, and culture but also their profound understanding of the school’s overall environment and development trends. Therefore, school organizational climate discussed in this paper is the “school organizational climate perceived by teachers”.

According to Bronfenbrenner’s [14] ecosystem theory, the school, which is the main working environment for teachers, is the core component of the microsystem that provides a platform for teachers to grow and develop. Therefore, the impact of the school environment on teachers is particularly crucial, and among these factors, school organizational climate plays a pivotal role. The organizational climate of a school not only shapes its overall appearance but also profoundly influences the behaviors and attitudes of its members, including teachers and students [15]. At the same time, social information processing theory also points out that the social environment in which individuals are located provides various pieces of information, and individuals’ perception of this environmental information is an important factor influencing their subsequent attitudes and behaviors [16]. Previous studies have demonstrated that the organizational climate plays a significant role in organizational citizenship behaviors [17]. Organizational justice in this climate has been shown to be an important dimension influencing organizational citizenship behavior, i.e., within organizations. When teachers feel they are treated fairly and with respect (i.e., when there is a strong sense of organizational justice), they are more likely to exhibit organizational citizenship behavior, e.g., voluntarily providing help and support to the organization and their colleagues [18]. In addition, organizational climate can either strengthen or weaken the process by which motivation influences proactive behavior [19]. According to the principles of social exchange and reciprocity, members who feel supported by their organization demonstrate organizational citizenship behaviors. Employees feel obliged to give back to the organization through positive behaviors when they are supported by the organization [20].

Based on this, we proposed the following hypothesis:

**H1:** *School organizational climate positively predicts teachers’ organizational citizenship behaviors*.

### 1.2. The Mediating Role of Teaching Efficacy

Teachers’ sense of teaching efficacy is a subjective belief and judgment regarding whether they can skillfully master teaching skills, successfully achieve their teaching tasks, and realize the established teaching goals. This reflects not only their understanding of education but also their confidence in their teaching abilities, which is a source of internal motivation that drives teachers to pursue excellence [21]. Teachers’ sense of teaching efficacy is closely associated with self-efficacy in a broader sense. Teachers’ sense of self-efficacy has been shown to positively predict their organizational citizenship behavior [22]. Moreover, Dussault [23] revealed a substantial positive association between teachers’ teaching efficacy and organizational citizenship behaviors, indicating that a stronger sense of teachers’ personal teaching efficacy contributed to more organizational citizenship behaviors, such as altruism, politeness, responsibility, and civic virtues, in the educational setting.

Bandura [24] proposed that self-efficacy is based on four sources of information: direct experience, alternative experience, verbal persuasion, and information about emotional and physiological states. The organizational climate of the school contains significant verbal persuasion and emotional arousal, which are important sources of efficacy [25]. Furthermore, social information processing theory [16] can also provide an explanation for this. The social information surrounding individuals offers social cues that help them interpret their work environment, and different interpretations of the work environment can lead to different attitudes among individuals. In a positive school climate, teachers have opportunities to collaborate with their colleagues and can inspire student engagement and academic achievement, which greatly enhances their confidence in teaching [26]. This process serves as a form of emotional arousal, which in turn enhances teachers’ sense of teaching efficacy [25].

In addition to theoretical support, several empirical studies have examined the role of school organizational climate in enhancing the sense of teaching efficacy. The innovation climate of a career field positively predicts employee innovation efficacy [27]. Similar research has also been conducted in the field of education, which indicates that school organizational climate can enhance teachers’ sense of teaching efficacy [28]. A study demonstrated that kindergarten organizational climate significantly and positively predicted teachers’ professional learning and that teacher collective efficacy partially mediated this relationship [29]. Moreover, teacher collaboration and teaching efficacy were independent and chained mediators in the relationship between distributed leadership and classroom quality, which exemplifies the role of teacher collaboration climate in predicting teaching efficacy [30]. In addition, a study of the influence of school environment on teachers’ professional development revealed that school climate was a significant positive predictor of teachers’ sense of teaching efficacy [31].

According to the theoretical framework proposed by Gist and Mitchell [32], self-efficacy in a specific domain is influenced by what they describe as “evaluations of individual and environmental resources or constraints” (p. 191). This notion primarily refers to the cognitive assessments that individuals make regarding external environmental elements, which play a crucial role in determining their perceived self-efficacy. This suggests that teaching efficacy mediates the relationship between cognitive evaluations of environmental factors (climate factors) and individual behaviors in this domain (organizational citizenship behaviors). Meanwhile, the motivational model proposed by [33] suggests that “the self serves as a mediator between the environment and behavior”. Under this theoretical framework, the sense of teaching efficacy, as a part of the “self”, acts as a bridge between the environment (school organizational climate) and behavior (organizational citizenship behaviors). In addition, the environment–psychological state behavior model proposed by Meyer and Allen [34] provides theoretical support for the relationships between school organizational climate, teaching efficacy, and teachers’ organizational citizenship behavior [35]. In this model, the environment (such as school organizational climate) can influence an individual’s psychological state (such as the sense of teaching efficacy), thereby driving their behavioral performance.

Therefore, we proposed the following hypothesis:

**H2:** *Teaching efficacy mediates the relationship between school organizational climate and teachers’ organizational citizenship behaviors*.

### 1.3. The Moderating Role of Optimistic Traits

Although the social environment can influence an individual’s behavior, this susceptibility varies from person to person [36]. Optimism refers to the positive expectation and anticipation of the outcomes of future events [37]. Based on the positive emotion extension and construction theory [38], optimistic employees positively interpret events at work and often have positive emotions. This broadens their attention span, making it easier for them to develop and accept new ideas and practices and show altruistic behaviors. This leads to positive outcomes, such as improved adaptability [39], and motivates teachers to engage in positive organizational behavior [40]. Ugwu and Igbende [41] found that employees’ optimism significantly and positively predicts organizational citizenship behavior. Teacher optimism significantly predicts individual and organizational citizenship behavior [42].

Furthermore, the interaction between optimistic traits and teaching efficacy affects teachers’ organizational citizenship behaviors. The resource conservation theory emphasizes that individuals with more resources are less vulnerable to resource loss and have a greater capacity to acquire more resources. The theory’s gain spiral refers to the greater capacity of individuals with sufficient resources to acquire more, and the result of their acquisition is a greater incremental increase in resources [43]. In other words, a specific initial positive factor (e.g., optimistic traits) can stimulate a subsequent set of positive behaviors, which, in turn, create a spiraling effect. Specifically, optimistic traits, such as the personality traits of individual teachers, can increase the positive effect of teaching efficacy on organizational citizenship behaviors.

Therefore, based on the value-added spiral effect and previous empirical research, we hypothesized that the influence of teaching efficacy on teachers’ organizational citizenship behavior varies with optimistic traits. Therefore, we proposed the following hypothesis:

**H3:** *Optimistic traits moderate the relationship between teaching efficacy and organizational citizenship behaviors*.

In summary, this study mainly relies on social information processing theory and integrates ecosystem theory, self-efficacy theory, the motivation model, and resource conservation theory to analyze the psychological mechanisms and individual differences in the complex relationship between school organizational climate and teachers’ organizational citizenship behaviors. By systematically combing and integrating diverse theoretical frameworks, this study provides a novel and comprehensive perspective on how school organizational climate subtly shapes teachers’ behaviors. This study focuses on teachers’ internal psychological mechanisms and their external behavioral responses, thereby deepening their theoretical understanding and broadening the boundaries of practical application in the field of educational psychology.

On this basis, this study proposes to construct a moderated mediation model. In the model, teaching efficacy is a bridge between school organizational climate and teachers’ organizational citizenship behaviors to reveal the internal logic of how the former influences the latter through this mediating variable. The optimism trait acts as a moderating variable in the second half of the pathway of this mediation model, adding new insights into understanding how individual differences moderate this relationship.

In addition, this study seeks to extend and enrich social information processing theory based on previous research. By incorporating the key element of personal traits into the theoretical model, especially by introducing the analysis of moderating variables in the classic pathway of “environment → attitude/behavior,” this study will deepen the understanding of social information processing theory and expand its scope of application, thus contributing to the further development and improvement of the theory.

The theoretical model is presented in Figure 1.

## 2. Materials and Methods

### 2.1. Participants

The survey was conducted in September 2023 in an area of Henan Province. Teachers from ten typical primary schools and ten equally representative middle schools within the province were selected as participants in the questionnaire survey. When selecting these schools, we thoroughly considered the urban–rural differences and the diversity of educational resource allocation to ensure that the selected schools could comprehensively reflect the actual educational environment within the province, thereby ensuring the breadth and representativeness of the sample. Within these schools, the principle of convenience sampling was adopted to distribute 666 questionnaires to teachers. After deleting questionnaires with missing answers, 500 valid questionnaires were obtained, with a validity rate of 75.08%. The participants comprised 226 men and 274 women. Among the participants, 90 had a teaching experience of 0–5 years, 107 had an experience of 5–10 years, 198 had an experience of 10–20 years, and 105 had an experience of >20 years. Furthermore, seven participants were second-grade primary teachers, 44 were first-grade primary teachers, 74 held senior titles in primary education, 5 held Title III in secondary education, 129 held Title II in secondary education, 135 held Title I in secondary education, 74 held senior titles in secondary education, and 32 held no title. The average age was 34.14 (*SD* = 7.13) years.

### 2.2. Procedure

After obtaining informed consent from the relevant teachers at the school, a professionally trained postgraduate student in psychology acted as the main examiner. The examiner explained the principle of anonymity and guidelines to the participants, ensured that the participants fully understood them, and then distributed the paper questionnaires, which were completed and collected immediately. The collected data were analyzed using SPSS 26.0 for descriptive statistics and correlation analysis, model 4 in PROCESS for the mediation analysis, and model 14 for the moderation analysis.

### 2.3. Ethical Considerations

Written informed consent was obtained from all participants before conducting the survey. This study was conducted in strict compliance with ethical guidelines and data protection principles. The participants’ privacy was fully respected. Participation was voluntary, and the participants could skip questions or withdraw from this study at any time. All collected data were strictly anonymized to eliminate the risk of disclosing personal information and safeguard the participants’ privacy. The data were stored in SPSS 26.0, and access to the data was restricted to researchers in the project team. Each researcher was required to strictly adhere to the data confidentiality agreement to ensure the legality and legitimacy of data use. This data collection activity was strictly reviewed and approved by the Ethics Committee of the school.

### 2.4. Measures

#### 2.4.1. Organizational Citizenship Behavior

Organizational citizenship behavior was measured using the Organizational Citizenship Behavior Questionnaire (OCBQ) developed by Farh et al. [44], which comprises 20 items in five dimensions: identification with the company, altruism toward colleagues, conscientiousness, interpersonal harmony, and protecting company resources. Responses were rated on a four-point Likert scale (1 = strongly disagree; 4 = strongly agree). Questions 4, 9, 14, 15, 16, 18, and 20 were reverse scored. Higher scores indicated stronger organizational citizenship behavior. The OCBQ had a Cronbach’s alpha coefficient of 0.92 in this study. The Cronbach’s alpha coefficient range for each dimension is 0.75 to 0.86, with a KMO value of 0.92. The results of Bartlett’s Test of Sphericity show a *p*-value < 0.001. The confirmatory factor analysis results indicate that χ^2^/df < 5, CFI = 0.90, RMSEA = 0.08 (90% CI = 0.08, 0.09), and SRMR = 0.06, suggesting good structural validity.

#### 2.4.2. School Organizational Climate

School organizational climate was measured using the School Organizational Climate Scale (SOCS) developed by Hart et al. [12], which comprises 11 items to evaluate appraisal and recognition, curriculum coordination, effective discipline policy, excessive work demands, goal congruence, participative decision making, professional growth, professional interaction, role clarity, student orientation, and supportive leadership. Responses were rated on a four-point Likert scale (1 = strongly disagree; 4 = strongly agree). Questions 4, 14, 15, 17, 18, 23, and 52 were reverse scored. Higher scores indicated a better school organizational climate. This signifies that teachers can perceive greater support and care from the leadership and represents a clear and profound understanding of their work goals and professional responsibilities. Furthermore, in such an atmosphere, teachers can support each other and communicate smoothly, and the school genuinely promotes teachers’ professional development and personal growth. The SOCS had a Cronbach’s alpha coefficient of 0.95 in this study. The Cronbach’s alpha coefficient range for each dimension is 0.71 to 0.85, with a KMO value of 0.95. The results of Bartlett’s Test of Sphericity show a *p*-value < 0.001. The confirmatory factor analysis results indicate that χ^2^/df < 4, RMSEA = 0.07 (90% CI = 0.06, 0.07), SRMR = 0.07, PGFI = 0.66, and PNFI = 0.67, suggesting good structural validity.

#### 2.4.3. Teaching Efficacy

Teaching efficacy was measured using the Teacher Academic Optimism Scale developed by Beard et al. [45], which comprises 11 items in three dimensions: teacher efficacy, trust in students and parents, and academic emphasis. Responses were rated on a four-point Likert scale (1 = strongly disagree; 4 = strongly agree). The scale had a Cronbach’s alpha coefficient of 0.86 in this study. The Cronbach’s alpha coefficient range for each dimension is 0.63 to 0.75, with a KMO value of 0.90. The results of Bartlett’s Test of Sphericity show a *p*-value < 0.001. The confirmatory factor analysis results indicate that χ^2^/df < 5, CFI = 0.91, GFI = 0.93, RMSEA = 0.09 (90% CI = 0.07, 0.10), and SRMR = 0.05, suggesting good structural validity.

#### 2.4.4. Optimistic Traits

Optimistic traits were measured using the Optimistic Traits Questionnaire developed by Jiang [46] by revising the Life Orientation Test (LOT) Scale developed by Scheieret et al. [47]. The reliability and validity of the questionnaire were tested in the Chinese cultural context with good results, with a Cronbach’s alpha coefficient of 0.68 and good structural and criterion-related validity. The questionnaire comprises ten items. Responses were rated on a four-point Likert scale (1 = strongly disagree; 4 = strongly agree). Questions 3, 7, and 9 were reverse scored. Higher scores indicated higher optimism. The scale had a Cronbach’s alpha coefficient of 0.68 in this study, which was acceptable. The KMO value is 0.78, and the results of Bartlett’s Test of Sphericity show a *p*-value < 0.001. The confirmatory factor analysis results indicate that χ^2^/df < 3, CFI = 0.91, GFI = 0.96, RMSEA = 0.06 (90% CI = 0.05, 0.07), and SRMR = 0.05, suggesting good structural validity.

## 3. Results

### 3.1. Common-Method Bias

As data acquisition relied on self-reporting, it potentially introduced the common method bias. To address this, Harman’s single-factor test was used to assess this bias. The outcome revealed 19 factors that explained 24.51% of the overall variance in the initial factors. This was below the critical threshold of 40%, suggesting that this study was devoid of significant common method bias issues.

### 3.2. Correlations Among Primary Variables

The optimistic traits, school organizational climate, teaching efficacy, and teacher organizational citizenship behavior variables were correlated, and the descriptive statistics and correlation coefficients are shown in Table 1. Two significant correlations were found among optimistic traits, school organizational climate, teaching efficacy, and teachers’ organizational citizenship behaviors.

### 3.3. An Analysis of the Mediating Role of Teaching Efficacy

PROCESS Macro was used with teaching age and title set as control variables. Model 4 was chosen based on templates to test for mediating effects (Figure 2). School organizational climate, teacher organizational citizenship behaviors, and teaching efficacy scores were standardized before testing. The results revealed that school organizational climate significantly and positively predicted teachers’ organizational citizenship behaviors (Table 2). When teaching efficacy was added, school organizational climate positively predicted teachers’ organizational citizenship behaviors through teaching efficacy.

The significance of the mediating effect was assessed using a bootstrap analysis. As shown in Table 3, teaching efficacy significantly mediated the relationship between school organizational climate and teachers’ organizational citizenship behaviors.

### 3.4. Moderating Effect of Optimistic Traits

Based on the research hypotheses and further focus on the effect of optimistic traits on teaching efficacy and teachers’ organizational citizenship behaviors, optimistic traits were used as moderating variables for in-depth analysis and exploration. Model 14 in PROCESS Macro was used to examine the mediating effect of optimistic traits. A significant interaction was observed between teaching efficacy and optimistic traits, which influenced teachers’ organizational citizenship behaviors (Table 4).

A simple slope analysis plot was constructed to visualize the moderating effect of optimistic traits (Figure 3). The results indicated that at low optimistic trait levels, teaching efficacy was not a significant predictor of teachers’ organizational citizenship behaviors; conversely, at high optimistic trait levels, teaching efficacy was a significant positive predictor of teachers’ organizational citizenship behaviors (*β* = 0.20, *p* < 0.001). As the moderating variables in this study were continuous, a simple slope test was conducted using the Johnson-Neyman (J-N) method. The results demonstrated the significant effect of teaching efficacy and optimistic traits on teachers’ organizational citizenship behavior (Figure 4). The optimistic trait score was >−0.63 (after standardization), and the confidence interval for the effect of teaching efficacy on teachers’ organizational citizenship behaviors did not contain 0, indicating that teaching efficacy had a significant effect on teachers’ organizational citizenship behaviors. As the optimistic trait scores increased, the effect also increased.

## 4. Discussion

This study demonstrated that teaching efficacy mediated the relationship between school organizational climate and teachers’ organizational citizenship behaviors and that optimistic traits moderated the second half of the pathway, verifying the proposed moderated mediation model.

### 4.1. School Organizational Climate and Teachers’ Organizational Citizenship Behaviors

School organizational climate significantly and positively predicted teachers’ organizational citizenship behaviors, supporting H1.

According to social information processing theory [16], individuals’ activities and behaviors do not occur in a vacuum but are typically influenced by complex situations. When schools cultivate a positive and supportive organizational climate, it makes teachers feel deeply respected and valued. Driven by such perceptions, teachers naturally tend to exhibit more organizational citizenship behavior. Additionally, a previous study revealed the existence of a social exchange relationship between teachers and universities, which involved realizing a community of values through interaction and cooperation between teachers and universities [40]. If teachers perceive a good atmosphere in their schools during a social exchange, they will likely exhibit positive and beneficial organizational citizenship behaviors, including school-friendly, proactive, and helping behaviors [40]. The results of this study provide empirical support for the social exchange relationships discussed above. The results indicated that positive and healthy school organizational climates provide teachers with a sense of psychological security and belonging. In this atmosphere, teachers will likely perceive themselves as integral members of the school family. Consequently, they become enthusiastic about participating in school activities and display organizational citizenship behaviors. They may take on extra workload, enthusiastically help their colleagues solve problems, and actively participate in the school’s decision-making processes.

### 4.2. Mediating Role of Teaching Efficacy

Perceptions of teaching efficacy mediated the relationship between school organizational climate and teachers’ organizational citizenship behaviors. First, school organizational climate significantly and positively predicted teachers’ sense of teaching efficacy, which was consistent with the findings of Nguyen et al. [48], who showed that a positive school climate significantly enhanced teachers’ sense of teaching efficacy.

The Job Demand-Resource (JD-R) model explores the intrinsic relationship between job resources and personal resources, suggesting that acquiring job resources promotes personal resource development [49]. When teachers perceive a stronger organizational climate, their job resource needs are met, facilitating their utilization of personal resources and enhancing their teaching efficacy. Furthermore, Ryan and Deci [50] proposed the self-determination theory (SDT), which categorizes needs as autonomy, competence, and relationship. Autonomy refers to the need for self-recognition and ownership of individual decision making and action, competence refers to the need for the ability to successfully solve life’s difficult problems, and relationship refers to the need for interdependent social resources to solve specific problems. The fulfillment of basic psychological needs enhances internal motivation [51]. Similarly, social information processing theory [16] emphasizes that the social context in which people work is a crucial source of information, having significant instructional and guiding effects on their attitudes. In the school ecosystem, when the organizational climate is harmonious and positive, teachers’ basic psychological needs are satisfied; thus, teachers have a strong interest and enthusiasm in teaching and devote energy and time to improving the quality and effectiveness of teaching. This positive engagement not only helps teachers achieve excellent teaching results but also significantly enhances their sense of teaching efficacy. This makes teachers more confident in facing educational challenges and promotes the improvement of the overall quality of education in schools.

Second, teachers’ teaching efficacy significantly and positively predicted their organizational citizenship behaviors, which was consistent with previous study findings. Teachers’ teaching efficacy, as their subjective judgment of their teaching ability, directly affects their teaching motivation and behavioral choices. When teachers evaluate their teaching competencies highly, they are inclined to respond positively to teaching challenges, take the initiative to assume responsibility, and seek opportunities for self-improvement. This positive mindset and behavioral pattern are highly compatible with the connotation of organizational citizenship behaviors, that is, voluntarily contributing beyond role responsibilities.

This study found that although the sense of teaching efficacy mediates between school organizational climate and teachers’ organizational citizenship behaviors, its relatively small effect size may imply that the influence of school organizational climate on teachers’ organizational citizenship behaviors is diverse and complex. This phenomenon can be deeply understood from the social information processing theory perspective, which emphasizes the potential shaping power of external social information on individual behavior. However, when constructing the model in this study, we focused only on the two core variables of school organizational climate and sense of teaching efficacy, neglecting other potentially important social information factors that may also be intertwined with school organizational climate and jointly influence teachers’ organizational citizenship behaviors (this also points the direction for future, more in-depth research to explore other possible influencing factors and suggests that future research can further expand the study sample).

Therefore, we further explore the potential role of a moderating variable (optimistic traits) in this relationship in the following text. It needs to be clarified that even with a relatively small mediation effect, the sense of teaching efficacy remains a crucial factor that cannot be ignored, as it connects school organizational climate and teachers’ organizational citizenship behaviors. This bridging role reveals how the school environment indirectly influences teachers’ external behavioral manifestations by affecting their internal psychological states.

### 4.3. Moderating Role of Optimistic Traits

Optimistic traits moderated the relationship between perceptions of teaching efficacy and teachers’ organizational citizenship behaviors. The predictive effect of teaching efficacy on teachers’ organizational citizenship behavior was only significant when the optimistic traits exceeded −0.63 (after standardization). In other words, strong optimistic traits enhanced the effect of a positive perception of teaching efficacy on organizational citizenship behaviors.

This threshold may represent a tipping point for the optimistic traits, where teachers’ optimism is sufficiently strong to contribute to their positive beliefs and expectations about teaching and learning. This suggests that optimistic traits are more than merely positive attitudes; they must reach a certain level to function synergistically with teaching efficacy. When optimistic traits exceed −0.63, teachers may be confident that they can influence students’ learning and development, and this confidence may translate into a higher sense of teaching efficacy. Increased teaching efficacy may motivate teachers to participate actively in school activities, including organizational citizenship behaviors that go beyond their regular duties. This may include volunteering to help colleagues and participating in charitable school events. Moreover, teaching efficacy is a source of motivation for teachers. This motivation may be strengthened when optimistic traits exceed a certain threshold, causing teachers to become proactive in pursuing their teaching goals and participating in school activities.

From the perspective of Trait Activation Theory [52], the findings of this study further confirm the process of personality trait expression in specific contexts. Optimistic traits, such as positive personality traits, are only activated and expressed in situations that are congruent with them. In this study, when the trait of optimism reaches or exceeds a certain threshold, it synergizes with the sense of teaching efficacy to jointly influence teachers’ organizational citizenship behaviors. A previous study found that positive emotions significantly affect innovative behavior, as positive emotions are pleasurable experiences that satisfy an individual’s needs and motivate positive behavior [53]. Thus, the combined effect of optimistic traits and teaching efficacy in the present study helped develop and maintain positive emotions, which in turn motivated individuals to engage in positive behaviors.

In summary, when teachers maintain a high sense of teaching efficacy due to their optimistic traits, they are likely to translate this positive state and belief into practical actions and exhibit organizational citizenship behaviors. For instance, they are likely to actively participate in school activities and develop close collaborative relationships with colleagues to jointly contribute to the overall interests of the school.

## 5. Conclusions

This study revealed that organizational climate significantly and positively predicted teachers’ organizational citizenship behavior. This indicated the role of the organizational environment in shaping individual behaviors and highlighted the importance of creating a positive work environment to stimulate teachers’ intrinsic motivation and sense of responsibility. In addition, this study demonstrated the mediating role of teaching efficacy in this relationship. When the organizational climate of a school is positive, teachers are likely to have high self-efficacy and believe that they can teach competently and positively impact their students. This positive psychological experience leads them to become engaged in school activities and display organizational citizenship behaviors. This result emphasizes the importance of enhancing teachers’ teaching efficacy in promoting all aspects of schools. Furthermore, this study found that optimistic traits moderated the relationship between teaching efficacy and teachers’ organizational citizenship behaviors. This finding suggests that teachers’ optimism enhances the contribution of teaching efficacy to their organizational citizenship behaviors. This provides new perspectives on how to promote school development by fostering teachers’ optimism.

This study had several limitations. First, the diversity of the sample and geographical differences may have affected the generalizability of the findings. In terms of sample diversity, if the study sample is too focused on one particular group, the findings may not fully reflect the diversity of relationships between teachers and the organizational climate in education. To enhance the generalizability of this study, efforts should be made to expand the sample in the future. Furthermore, geographical differences may also exert an impact on the relationship between organizational climate and teacher behavior. Schools in different regions differ in terms of educational policies, resource allocation, cultural traditions, and socio-economic environments, which may lead to different organizational climates, which in turn may affect teachers’ behaviors. To overcome this limitation, future surveys should be conducted across geographical boundaries in several representative districts to provide richer and more specific guidance for educational practice. Second, as this study used a cross-sectional questionnaire survey method, causal relationships between variables could not be identified. In the future, longitudinal studies should be conducted to reveal the causal relationship between these variables.

Meanwhile, to fully comprehend the manifestations and impacts of school climate and organizational citizenship behavior across diverse educational settings, future research must delve deeper into cross-cultural comparisons. This endeavor will not only assist in validating the universality and specificity of current research findings but also unveil the unique interconnections between school climate and individual behavior within various cultural contexts. By engaging in cross-cultural research, we can acquire more abundant and diversified data, enabling us to propose educational strategies and recommendations with a broader global perspective. Moreover, to gain a more comprehensive and in-depth understanding of the core topic of this study and effectively compensate for the possible limitations of the questionnaire, qualitative research methods should be integrated into the follow-up research process. While quantitative research provides macro results of data analysis, qualitative research can provide specific situational analyses. Therefore, follow-up studies should select representative teachers, school leaders, or students to conduct interviews around issues that could not be explored in depth in the questionnaire, such as behavioral motivation and decision-making process, to ascertain their views on the organizational climate, teachers’ behaviors, and their interrelationships.

Our findings suggest that school administrators should promote positive organizational climates in their schools by establishing regulations that allow everyone to contribute to the school climate. This implies that the regulations should not only ensure the orderly operation of the school on a daily basis but also reflect respect and inclusiveness for the individual differences among teachers and students, providing them with diversified participation channels and expression platforms. Such regulations can stimulate the sense of belonging and responsibility of all teachers and students, jointly creating a positive, harmonious, and collectively progressive organizational climate. Additionally, they can strengthen the cultivation of teachers’ sense of personal efficacy and optimism by establishing diversified evaluation systems, encouraging continued education, and organizing professional training. By verifying the mediating role of teaching efficacy and the moderating role of optimistic traits, this study deepens the understanding of the interaction mechanism between individual behavior and organizational environment. It also expands the research perspective from general organizations to the educational field, especially for the special occupational group of teachers, enriching the research content of organizational citizenship behavior in the educational field.

Furthermore, the core theoretical contribution of this study lies in deepening and broadening the content and scope of social information processing theory by integrating the crucial element of individual traits into the theoretical model, thereby providing new insights for subsequent research. In addition, the findings can guide school management practices, help build a more harmonious and efficient educational environment and promote teachers’ professional development by focusing on teachers’ psychological health and professional development needs, which helps teachers improve their sense of teaching efficacy and optimistic traits, promoting the emergence of their organizational citizenship behaviors and improving the overall quality and effectiveness of teaching and learning.

## Figures and Tables

**Figure 1 behavsci-14-01130-f001:**
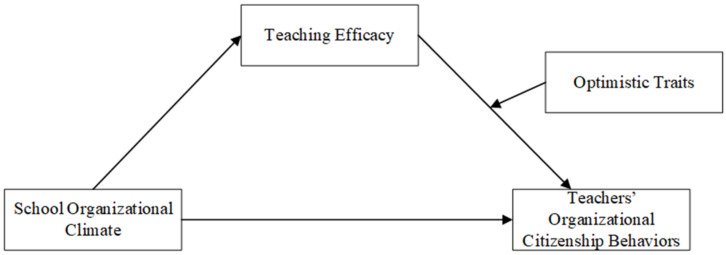
Hypothesized model.

**Figure 2 behavsci-14-01130-f002:**
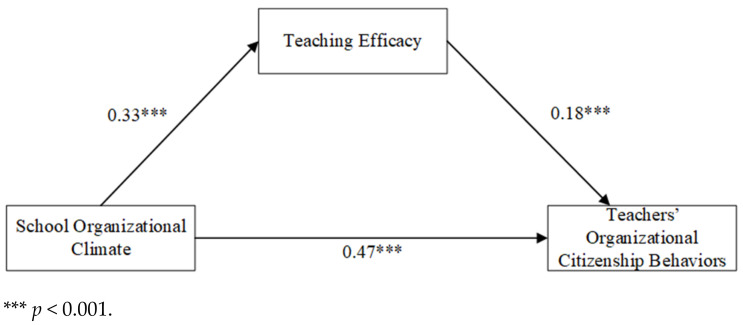
Intermediary model diagram.

**Figure 3 behavsci-14-01130-f003:**
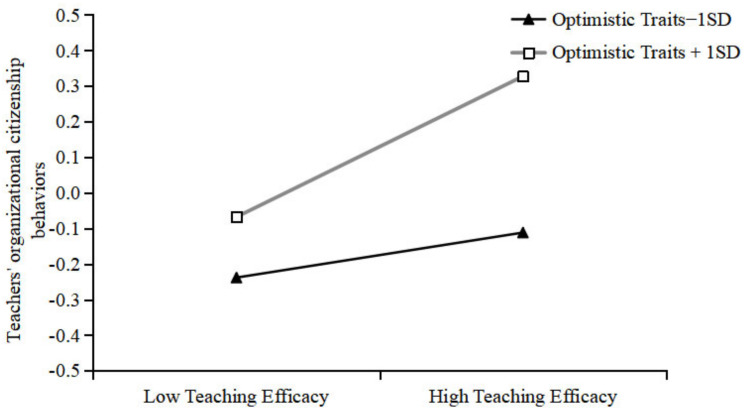
Simple slope test.

**Figure 4 behavsci-14-01130-f004:**
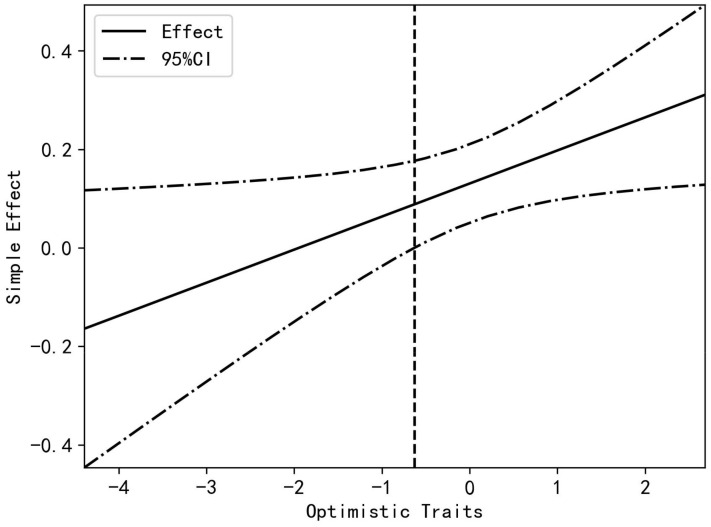
J-N simple slope analysis.

**Table 1 behavsci-14-01130-t001:** Correlations among variables.

	*M*	*SD*	1	2	3	4
1. Optimistic Traits	29.76	3.82	1			
2. School Organizational Climate	146.95	21.45	0.28 ***	1		
3. Teaching Efficacy	36.27	4.48	0.34 ***	0.33 ***	1	
4. Teachers’ Organizational Citizenship Behaviors	65.54	8.60	0.32 ***	0.54 ***	0.34 ***	1

*** *p* < 0.001.

**Table 2 behavsci-14-01130-t002:** Regression analysis results of the intermediation model.

Regression Equation		Model Fit			Regression Coefficients and Significance		
Outcome Variable	Predictor Variable	*R*	*R* ^2^	*F*	*β*	*SE*	*t*
Teachers’ Organizational Citizenship Behaviors	School Organizational Climate	0.54	0.29	69.00 ***	0.53	0.04	13.79 ***
Teaching Efficacy	School Organizational Climate	0.34	0.12	21.76 ***	0.33	0.04	7.57 ***
Teachers’ Organizational Citizenship Behaviors	School Organizational Climate	0.57	0.32	58.69 ***	0.47	0.04	11.87 ***
	Teaching efficacy				0.18	0.04	4.46 ***

*** *p* < 0.001.

**Table 3 behavsci-14-01130-t003:** Results of the mediating effect test.

	Efficiency Value	Boot SE	95%CI
Total Effect	0.53	0.04	[0.46–0.61]
Direct Effect	0.47	0.04	[0.40–0.55]
Indirect Effect	0.06	0.02	[0.03–0.10]

**Table 4 behavsci-14-01130-t004:** Moderating effect test of optimistic traits.

Regression Equation		Model Fit			Regression Coefficients and Significance			95%CI	
Outcome Variable	Predictor Variable	*R*	*R* ^2^	*F*	*β*	*SE*	*t*	LLCI	ULCI
Teachers’ Organizational Citizenship Behaviors	School Organizational Climate	0.60	0.35	43.77 ***	0.44	0.04	10.88 ***	0.36	0.52
	Teaching Efficacy				0.13	0.04	3.23 **	0.05	0.21
	Optimistic Traits				0.15	0.04	3.81 ***	0.07	0.23
	Teaching Efficacy × Optimistic Traits				0.07	0.03	2.15 *	0.006	0.13

*** *p* < 0.001., ** *p* < 0.01, * *p* < 0.05.

## Data Availability

The raw data supporting the conclusions of this article will be made available by the authors upon request.

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
