# Peer review of "The Relationship Between School Organizational Climate and Teachers’ Organizational Citizenship Behaviors: The Mediating Role of Teaching Efficacy and Moderating Role of Optimistic Traits"

_behavsci, 2024, doi:10.3390/bs14121130_

Round 1

Reviewer 1 Report

Comments and Suggestions for Authors

Contextualization and Literature Review: While the paper effectively uses established theories (e.g., ecological and social exchange theories) to frame its hypotheses, incorporating more recent and diverse studies, especially from international contexts, could strengthen the literature review. This would enhance the study's relevance and contextual depth.

Discussion of Findings: The discussion section could be expanded to delve deeper into the implications of the findings on teaching efficacy and optimistic traits in educational settings. Highlighting the practical applications of these findings for school administrators and policy-makers would provide additional value for readers.

Theoretical Contributions: The paper builds on existing research, but the originality could be enhanced by discussing how these findings provide new insights or perspectives on school climate and organizational citizenship behavior, especially in educational psychology. Emphasizing the unique aspects of this study’s contribution would help to position it within the broader field.

Presentation of Results: The results are well-organized and supported with figures and tables, which aids in clarity. However, providing a summary table that briefly outlines the key findings in relation to each hypothesis would improve accessibility for readers.

Implications for Practice: The conclusion could be strengthened by explicitly addressing how school administrators and educators can apply these insights. For instance, practical strategies to foster optimistic traits and teaching efficacy within school environments would make the findings more actionable.

Limitations and Future Research: The limitations section is well-noted, particularly regarding the need for longitudinal studies and qualitative methods. Future studies could also consider cross-cultural comparisons to assess the generalizability of the findings in different educational contexts.

Language and Clarity: Overall, the language is clear and professional, though minor grammatical refinements could further improve readability. Ensuring consistent terminology throughout the paper will help maintain clarity, especially when discussing concepts like "organizational climate" and "citizenship behaviors."

Author Response

Dear reviewer,please see the attachment.

Reviewer 2 Report

Comments and Suggestions for Authors

The paper has interesting points about important issues.

However, I would like to do some remarks.

First, the author must highlight, soon in the abstract and introduction, the paper conceptual contribution. That is not clear yet.

About the scales and the measurement model, some constructs are high-level ones. They have dimensions. However, they are modelled as low order. Is that correct? If they are high order constructs, the reliability (alpha de Cronbach) should be assessed for its dimensions - low order construct. If they are low order, they does not have dimensions.

Besides, the paper does not the validity of the scales - convergent and discriminant

Author Response

(The authors gave the same response as above.)

Reviewer 3 Report

Comments and Suggestions for Authors

This study addresses a timely and important topic, considering the significant impact that teachers' skills, personal characteristics, and their working environment have on the quality of education. However, as an independent piece of research, this paper presents multiple critical issues that must be resolved before it can be deemed suitable for publication in a prestigious academic journal that emphasizes scientific rigor in its evaluation of research questions.

To begin with, there is a fundamental issue concerning the level of analysis. The primary independent variable, school organizational climate, is intended to be an organizational-level variable. Yet, the mediating, moderating, and dependent variables are positioned at the individual level. Upon reviewing the data collection method, it is clear that a self-report questionnaire was employed. This approach is inadequate for capturing organizational-level variables because it measures the respondents' perceptions of the organizational climate rather than the climate itself. Accurate measurement of organizational climate requires alternative data collection methods, as self-reported data introduces a bias. Individuals' perceptions are likely influenced by their personal characteristics, such as optimism. For instance, teachers with optimistic traits might view their work environment more positively, and those with high teaching efficacy may also report a favorable organizational climate. A comprehensive redesign of the data collection strategy is necessary to accurately capture the true organizational climate, rather than subjective perceptions.

Moreover, the manuscript lacks a robust theoretical framework. While the authors reference existing theories, they do not provide their own logical reasoning for the hypothesized relationships between variables. This reliance on external theories without offering original insights constitutes a well-known logical fallacy. For example, the proposed interaction effect of teaching efficacy and optimistic traits on organizational citizenship behavior is not convincingly argued. One might contend that effective teachers might focus on their own tasks to capitalize on their strengths rather than engage in citizenship behavior. Similarly, optimism might be self-directed rather than extending to others. Without addressing these criticisms, the hypotheses remain weak. Furthermore, the manuscript does not adequately explain why teaching efficacy should serve as a mediating variable if the organizational climate directly influences citizenship behavior. Significant revisions are needed to develop strong, original theories that can underpin the hypotheses and make them persuasive.

Additionally, the concept of organizational citizenship behavior has evolved significantly since its inception in the 1980s. It now encompasses at least two sub-dimensions: citizenship behavior towards the organization and towards individuals, such as colleagues. This study does not reflect this nuanced understanding, indicating a limited literature review. A thorough examination of the existing body of knowledge is essential to accurately capture the complexity and dimensions of organizational citizenship behavior.

Moreover, the sampling procedure used in this study is fundamentally flawed as it violates the principle of representativeness. The manuscript does not clearly define the target population, whether it is teachers in general, teachers in China, or teachers in a specific region of China. The use of convenience sampling is apparent, which undermines the study's scientific validity. Without a representative sample, the analyses lack meaning, as they are based on incorrect data. Consequently, the current findings are unreliable. For future research, it is essential to ensure that data collection methods are designed to accurately represent the target population.

Furthermore, the manuscript lacks clarity in articulating the research objectives and questions. A well-defined research question provides a clear focus and direction for the study. The authors must ensure that the research objectives are explicitly stated and aligned with the hypotheses and data analysis. This alignment is crucial for establishing a coherent narrative throughout the manuscript.

In conclusion, this manuscript, in its current form, has significant shortcomings as an independent research study. To meet the standards of a journal such as Behavioral Sciences, which values scientific rigor, the manuscript requires a complete revision from beginning to end. I hope these comments assist the authors in finding a clear path to significantly improve their study for future publication consideration. My intention is to provide constructive feedback that will guide the authors towards enhancing the quality and impact of their research.

Comments on the Quality of English Language

The quality of English does not limit my understanding of the research, but the English could be improved to more clearly express the research.

Author Response

(The authors gave the same response as above.)

Round 2

Reviewer 2 Report

Comments and Suggestions for Authors

The paper has improved since the later version

Reviewer 3 Report

Comments and Suggestions for Authors

No further comments.

Comments on the Quality of English Language

 The quality of English does not limit my understanding of the research, but the English could be improved to more clearly express the research.